

# The Far-INfrarEd Spectrometer for Surface Emissivity (FINESSE) Part II: First measurements of the emissivity of water in the far-infrared.

Laura Warwick[1], Jonathan Murray[2,3], and Helen Brindley[2,3]

[1]ESA-ESTEC, Noordwijk, Netherlands
[2]Imperial College London, London, UK
[3]National Centre for Earth Observation, UK

**Correspondence:** Laura Warwick (laura.warwick@esa.int)

**Abstract.** In this paper we describe a method for retrieving surface emissivity across the wavenumber range 400-1600 cm[-1] using novel radiance measurements from the Far INfrarEd Spectrometer for Surface Emissivity (FINESSE) instrument. FINESSE is described in detail in part I of this paper. We apply the method to two sets of measurements of distilled water. The first set of emissivity retrievals is of distilled water heated above ambient temperature to enhance the signal to noise ratio. The

5 second set of emissivity retrievals is of ambient temperate water at a range of viewing angles. In both cases the observations agree well with calculations based on compiled refractive indices across the mid and far-infrared. It is found that the reduced contrast between the up and downwelling radiation in the ambient temperature case degrades the performance of the retrieval. Therefore a filter is developed to target regions of high contrast which improves the agreement between the ambient temperature emissivity retrieval and the predicted emissivity. These retrievals are, to the best of our knowledge, the first published

retrievals of the emissivity of water that extend into the far-infrared and demonstrate a method that can be used for the in-situ retrieval of the emissivity of other surfaces in the field.

## 1 Introduction

The emissivity of a surface is the ratio between the electromagnetic radiation emitted by that surface and the electromagnetic radiation emitted by a blackbody at the same temperature. Emissivity is a spectrally varying property that depends on many

factors including surface composition, roughness, temperature, and viewing angle (Li et al., 2013). Accurate knowledge of the Earth's surface emissivity in the infrared, particularly across the atmospheric window (833 to 1250 cm[-1], 8 to 12 μm), is essential for our understanding of the top of the atmosphere energy budget because it influences how the planet radiatively cools to space. The surface emissivity is also vital for determining the surface energy budget which governs the exchanges of heat and water at the surface (Liang et al., 2019).

The Earth's surface emissivity is well known in the atmospheric window through retrievals from many previous in-situ and satellite measurements. For example the Advanced Spaceborne Thermal Emission Reflection Radiometer (ASTER) Global Emissivity Dataset, compiled from data from the ASTER instrument on board NASA's Terra satellite, covers the globe at a



resolution of 100 m (Hulley et al., 2015). There are also libraries of spectrally resolved emissivity values for a range of surface types, for example the ECOsystem and Spaceborne Thermal Radiometer Experiment on Space Station (ECOSTRESS) spectral

library which comprises laboratory measurements of over 3000 material samples including soil, vegetation, water, man-made surfaces and lunar materials (Meerdink et al., 2019). The measurements in the ECOSTRESS library are available over a variety of spectral ranges however the ECOSTRESS library and other emissivity libraries do not extend into the far-infrared due to a lack of emissivity measurements at wavenumbers below 667 cm$^{-1}$ (wavelengths greater than 15 µm). This lack of measurements is of particular concern as theoretical studies have shown that including spectrally resolved far-infrared emissivity values in

radiative transfer calculations can have a discernible impact on the outgoing longwave radiation (Feldman et al., 2014; Huang et al., 2016) and thus affect estimates of the Earth's top of the atmosphere energy balance.

There are only a handful of existing far-infrared surface emissivity measurements and most of these are of are of snow and ice surfaces. These include two sets of measurements from ground based instruments and a set of airborne measurements. One set of ground based measurements was carried out using the Far-Infrared Radiation Mobile Observation System (FIRMOS)

instrument as a small part of a large winter campaign at the Zugspitze Observatory in the German Alps (Palchetti et al., 2020a). Measurements of surface emissivity were made for several snow types and were accompanied by characterisation of the snow surfaces. The results show that the emissivity varies with the snow properties. This is expected given that the measured emissivity of snow in the mid-infrared varies with snow properties (Hori et al., 2006) and modeling indicates that this also occurs in the far-infrared (Chen et al., 2014). However the viewing geometry of FIRMOS made these measurements

difficult to undertake as the instrument is designed for zenith and nadir measurements rather than the slant paths required to minimise the effects of instrument self emission during in-situ emissivity retrievals. Ground-based measurements of ice, in the far-infrared have also been reported by Borbas et al. (2021) using the Absolute Radiance Interferometer. This paper also reports measurements of the emissivity of pine needles and sand. Finally, retrievals of snow and ice emissivity were carried out using airborne measurements made by the Tropospheric Airborne Fourier Transform Spectrometer. This was done using

data from both a low altitude flight over the Greenland ice sheet (Bellisario et al., 2017) and from a high altitude flight over roughly the same area (Murray et al., 2020). The results from the low and high altitude flights agree within the measurement uncertainty and demonstrate the potential for the measurement of far-infrared emissivity from a satellite platform. This is further supported by theoretical studies in preparation for two far-infrared satellite missions, NASA's Polar Radiant Energy in the Far Infrared Experiment (PREFIRE) (L'Ecuyer et al., 2021) and the European Space Agency's Far Infrared Outgoing

Radiation Understanding and Monitoring (FORUM) mission (Palchetti et al., 2020b), which have demonstrated successful retrievals of far-infrared surface emissivity in cloud-free conditions with low total column water vapour (Xie et al., 2022; Ben-Yami et al., 2022). The need for good knowledge of the a-priori surface emissivity was also highlighted.

Overall, there is a clear need for further measurements of surface emissivity that extend into the far-infrared. In order to address this gap in measurements and provide a-priori knowledge for emissivity retrievals across the far-infrared, the Far IN-

55 frarEd Spectrometer for Surface Emissivity (FINESSE) has been developed at Imperial College London. FINESSE is designed to make in-situ measurements of emissivity in the wavenumber range 400 - 1600 cm$^{-1}$ (6.3 to 25 µm). The instrument is portable and able to view scenes over a continuous range of angles from nadir to zenith. A detailed description of the instru-





ment in provided in Part I of this paper (Murray et al., 2023). In this part II we demonstrate how FINESSE can be used to retrieve surface emissivity, focusing specifically on distilled water. In the next section we describe our retrieval approach and discuss how the the emissivity of distilled water can be theoretically modelled. Next we describe the two sets of measurements used in this study, the first using a heated water surface to enhance the signal to noise ratio and the second investigating angular dependence. The results section details the emissivities retrieved from these measurements. In both cases we consider the various sources of error and make comparison to the corresponding theoretical calculations. Conclusions are drawn in the final section.

## 2 Methodology

### 2.1 Emissivity Retrieval

We have chosen to adapt an emissivity retrieval method that has been successfully applied for the retrieval of surface emissivity in the mid-infrared using high resolution spectra (Newman et al., 2005). This method was used to retrieve surface emissivity of water at different temperatures and salinities using observations from a mid-infrared interferometer both from an aircraft and in the lab and has been adapted for the retrieval of far-infrared snow and ice emissivity from aircraft measurements (Bellisario et al., 2017). The method involves the measurement of upwelling radiation from the surface under study, as well as direct measurement of the downwelling radiation and determination of the transmission between the surface under study and the detector. Figure 1 shows the geometry of this measurement technique. Selecting this method allows us to make in-situ measurements without disturbing the surface being measured.

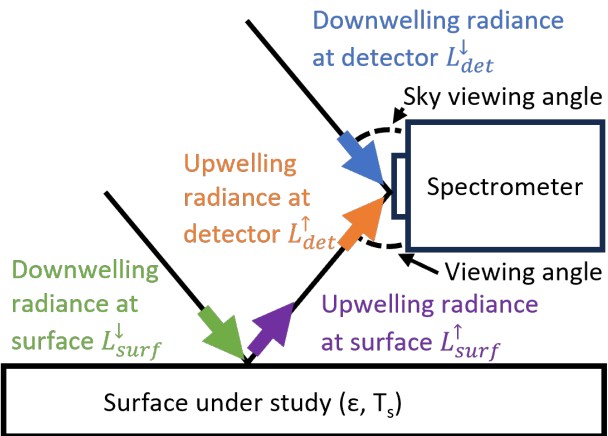

**Figure 1.** A schematic showing the geometry of the emissivity retrieval method.

Following the derivation by Newman et al. (2005), the upwelling radiance from a surface, $L^\uparrow_{surf}$, which is a function of viewing angle, $\theta$, is composed of a thermal emission term and a component of reflected radiation both of which are dependent





on the emissivity, $\epsilon$,

$$L^{\uparrow}_{surf}(\theta) = \epsilon(\theta)B(T_s) + (1 - \epsilon(\theta))L^{\downarrow}_{surf}(\theta), \tag{1}$$

where $L^{\downarrow}_{surf}$ is the downwelling radiation at the surface and $B(T_s)$ is the Planck function at surface temperature $T_s$. $L^{\downarrow}_{surf}$ and

$\epsilon$ are both functions of viewing angle and all values are spectrally dependant. However, when measuring surface emissivity, we must also consider the absorption and emission of the atmosphere between the surface and the detector. Therefore, the upwelling radiance measured by the detector $L^{\uparrow}_{det}(\theta)$ becomes,

$$L^{\uparrow}_{det}(\theta) = \tau(\theta)L^{\uparrow}_{surf}(\theta) + E^{\uparrow}(\theta), \tag{2}$$

where $\tau$ is the transmission and $E^{\uparrow}(\theta)$ the upwelling emission of the atmospheric layer between the surface and the spectrom-

eter. Similarly, the downwelling radiation at the surface differs from that measured by the spectrometer,

$$L^{\downarrow}_{surf}(\theta) = \tau(\theta)L^{\downarrow}_{det}(\theta) + E^{\downarrow}(\theta), \tag{3}$$

where $E^{\downarrow}(\theta)$ is the downwelling emission of the layer between the detector and the surface. During our in-situ measurements the distance between the spectrometer and the surface is at most a few metres, so for these distances we assume that the atmosphere is homogeneous and isothermal, in which case the up- and downwelling emission is the same and can be written

as,

$$(1 - \tau(\theta))B(T_a) \tag{4}$$

where $B(T_a)$ is the Planck function at the average temperature of the atmospheric layer, $T_a$. By rearranging the above equations, we find the following expression for surface emissivity,

$$\epsilon = \frac{L^{\uparrow}_{det} - \tau^2 L^{\downarrow}_{det} - (1 - \tau^2)B(T_a)}{\tau\{B(T_s) - \tau L^{\downarrow}_{det} - (1 - \tau)B(T_a)\}}. \tag{5}$$

Accurately and precisely determining the surface skin temperature is a vital part of the retrieval process. Numerous algorithms have been developed to do this when determining emissivity in the mid-infrared (e.g., Salvaggio and Miller, 2001). However many of these techniques make the assumption that the transmission between the surface and the detector is 1, which is not appropriate for the far-infrared where there is strong water vapour absorption. Therefore in this study we use the surface temperature retrieval method described by Newman et al. (2005). This method relies on the spectral smoothness of the surface

emission but takes into account the atmospheric path between the surface and the instrument. This is done by noting that $L^{\uparrow}_{surf}$ is composed of emitted surface radiation, which is a smoothly varying function, and reflected sky radiation which has spectral features. Therefore it should be possible to find a value for the reflectance, $\rho$, such that

$$L^{\uparrow}_{surf} - \rho L^{\downarrow}_{surf} = \epsilon B(T_s) \tag{6}$$





is a smooth function. By splitting equation 6 into small wavenumber intervals $\rho$ can be taken as constant within each wavenumber interval. Substituting in definitions from equations 2, 3 and 4, equation 6 can be rewritten as

$$\frac{1}{\tau}\{L_{det}^{\uparrow} - (1-\tau)B(T_a)\} - \rho\{\tau L_{det}^{\downarrow} + (1-\tau)B(T_a)\} = \epsilon B(T_s). \tag{7}$$

As this equation should still represent a smooth function, the value of $\rho$ can be altered in each wavenumber interval to minimise the root mean square difference between the left hand side of the equation and a quadratic fit of the left hand side of the equation. The average emissivity for the wavenumber band is $1-\rho$. Therefore the right hand side of the equation can be used to determine the surface temperature by inverting the Planck function. The retrieved temperature is then taken as the average of the retrieved temperature in each wavenumber interval. We choose to retrieve the surface temperature in the wavenumber range 800 to 1200 cm⁻¹ using intervals with a width of 40 cm⁻¹. This is to minimise errors cause by mischaracterisation of the atmosphere in between the surface and the instrument.

## 2.2 Modelling of Emissivity

The emissivity of surfaces that act as specular reflectors, such as water or ice, can be modelled using Fresnel equations. This relies on the knowledge of the complex refractive indices of the material (Masuda et al., 1988). The reflectance, $\rho$, can be expressed as a function of viewing angle and refractive index

$$\rho(n,\theta) = \frac{\left(|\gamma_{\parallel}|^2 + |\gamma_{\perp}|^2\right)^2}{2}, \tag{8}$$

where $n$ is the complex refractive index, $\theta$ is the viewing angle and $\gamma_{\parallel}$ and $\gamma_{\perp}$ are the complex reflectances polarized parallel and perpendicular to the plane of incidence and are given by

$$\gamma_{\parallel} = -\frac{n\cos\theta - \cos\theta'}{n\cos\theta + \cos\theta'}, \tag{9}$$

$$\gamma_{\perp} = \frac{\cos\theta - n\cos\theta'}{\cos\theta + n\cos\theta'}, \tag{10}$$

where $\theta'$ is the angle of refraction and is related to $\theta$ by

$$\sin\theta' = \frac{1}{n}\sin\theta. \tag{11}$$

We choose to use distilled water to develop and demonstrate the first emissivity retrievals using FINESSE. To the best of our knowledge there are no published values of the emissivity of distilled, fresh or sea-water that extend into the far-infrared. Furthermore the performance of the retrieval can be assessed through comparison to Fresnel calculations carried out

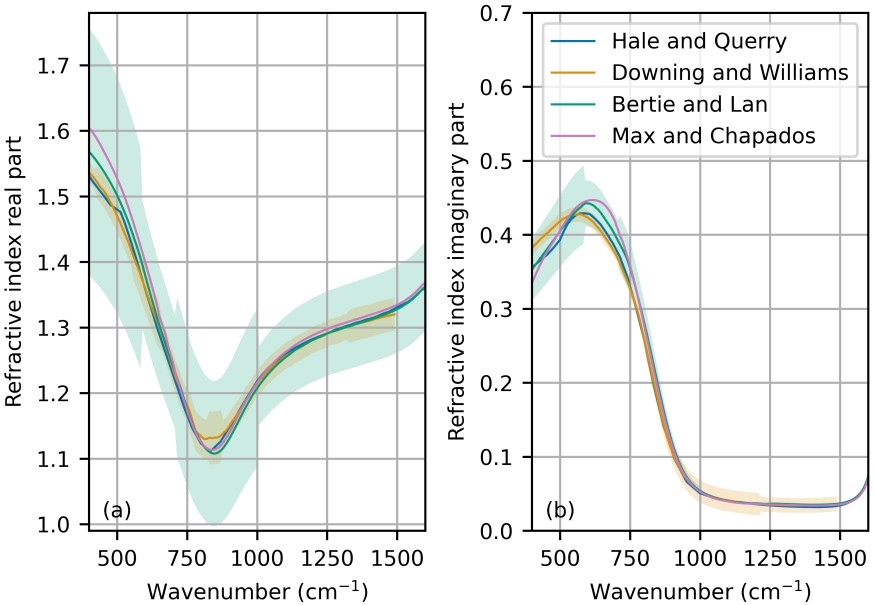

**Figure 2.** (a) real and (b) imaginary refractive indices from Hale and Querry (1973), Downing and Williams (1975), Bertie and Lan (1996) and Max and Chapados (2009). The orange and blue shading represents the quoted uncertainty values from Downing and Williams (1975) and Bertie and Lan (1996), respectively.

using published refractive index measurements. These refractive index measurements have been compiled by various sources
including Hale and Querry (1973), Downing and Williams (1975), Bertie and Lan (1996) and Max and Chapados (2009).
Figure 2 shows the real and imaginary parts of these refractive indices for a temperature of 298±2 K. The differences between
the different compilations are larger for the real part of the refractive index, particularly below 600 cm$^{-1}$. For the imaginary
part, the largest differences are seen below 800 cm$^{-1}$. These compilations use data acquired using a range of measurement
techniques that are distinct from our emissivity measurements. Uncertainty values for these refractive indices are only given by
Downing and Williams (1975) and Bertie and Lan (1996). These uncertainty values are shown as the orange and blue shading
around the respective lines. All values agree within the uncertainty values of the Bertie and Lan (1996) compilation.

## 3   The Measurements Undertaken

FINESSE was installed on the rooftop of Imperial College on the 11[th] February and 17[th] March 2022 under clear sky condi-
tions (Warwick et al., 2024). On the the 11[th] February measurements took place between 0930 and 1030 UTC. Over the course
of the hour, the ambient air temperature increased from 278 to 280.5 K and the relative humidity fluctuated between 75% and
65% with a general downward trend (figure 3a). A Grant Instruments T100 water bath was filled with distilled water and placed
in the field of view of FINESSE. The surface of the water was 22.6 cm below the height of the FINESSE pointing mirror. Prior



to the radiance measurements starting, the water was heated to a temperature of 303 K using the water bath. The water bath was then turned off and the water was allowed to naturally cool until all visible mist had dissipated from the surface of the water.

The water then continued to cool from a temperature of roughly 294 to 290 K over the course of the hour. FINESSE made radiance measurements in the pre-programmed measurement sequence: hot blackbody (HBB), ambient blackbody (ABB), water surface at 45°, HBB, ABB, water surface at 45°, HBB, ABB, clear sky at 135°. Figure 4a shows a photograph of the setup.

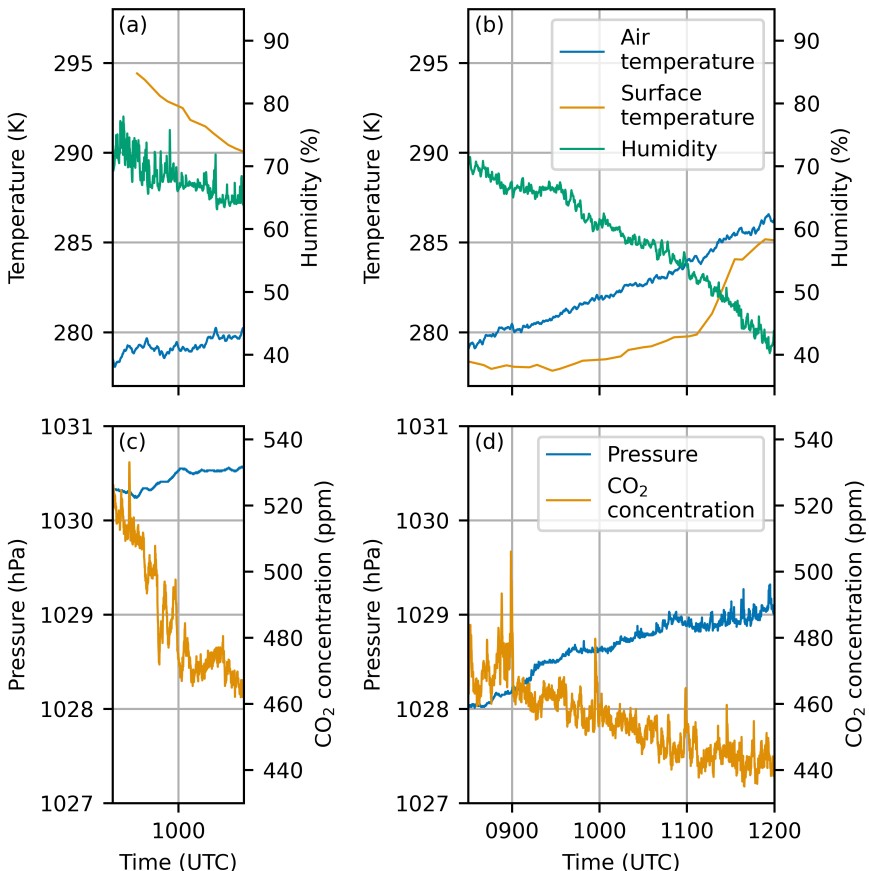

**Figure 3.** (a) The air temperature, water surface temperature and humidity during the 11[th] February measurements. (c) The measured pressure and $CO_2$ concentration during the measurements on the 11[th] February. (b and d) the same as (a and c) but for the 17[th] March.

The 17[th] March measurements took place over 3 hours 30 minutes between 0830 and 1200 UTC. Over the course of the measurements, the ambient atmospheric temperature increased from 278 to 287 K and the relative humidity decreased from 70

to 40%, see figure 3c. The distilled water was held in a long tray 28.3 cm below the pointing mirror. Measurements were made of the water surface at angles of 50°, 60° and 70 °, accompanied by measurements of the calibration targets and views of the downwelling radiance following the same sequence as the February measurements. The 50° measurements were undertaken





between 0855 and 0950 UTC, the 60° measurements between 1000 and 1050 UTC and the 70° measurements between 1100 and 1155. Figure 4b shows a photograph of the setup.

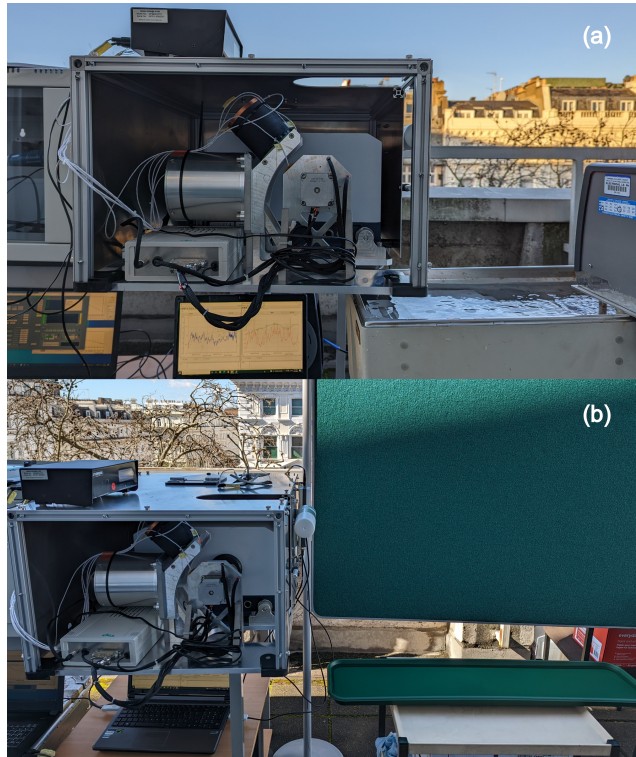

**Figure 4.** Photograph of the setup for FINESSE to make measurements of water emissivity on (a) 11[th] February and (b) 17[th] March. The ripples seen on the surface of the water bath in (a) are caused by the heating action of the water bath and were not present when the measurements were made. The green screen in figure (b) was used to shield the water surface from the breeze, again to ensure the surface was as flat as possible.

## 4 Results

### 4.1 Emissivity retrieval 11[th] February

Figure 5a shows the average spectra recorded for the downwelling sky view and upwelling water view over the course of the measurements. The upwelling spectrum is higher than the downwelling spectrum across the whole spectral range. This contrast is because the water was heated above the ambient atmospheric temperature. Figures 5b and c show the uncertainty on the measured up- and downwelling spectra due to uncertainty in the temperature and emissivity of the FINESSE blackbody cavities and the FINESSE noise equivalent spectral radiance (NESR). These were calculated using the methods described in Murray et al. (2023).



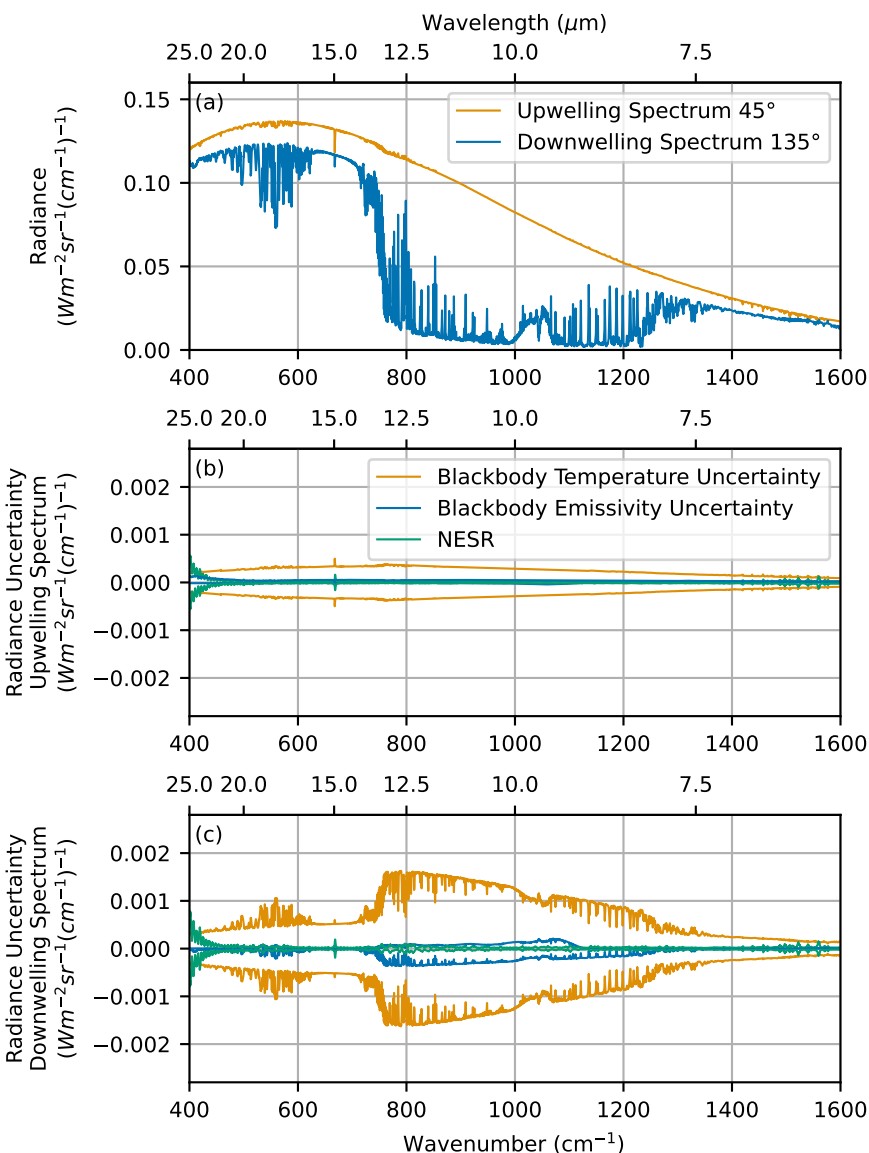

**Figure 5.** (a) Average up- and downwelling spectra measured on the 11[th] February. (b) Average uncertainty on the upwelling spectra calculated using the methods from Murray et al. (2023). (c) Average uncertainty on the downwelling spectra.



The transmission through the atmospheric path between the water surface and FINESSE was calculated using version 12.10 of the Line by Line Radiative Transfer Model (LBLRTM) (Clough et al., 2005). A separate transmission was calculated for each water view made by FINESSE. Values for temperature, pressure, humidity and $CO_2$ concentration were taken from FINESSE's auxiliary measurements (figure 3). The path length was calculated using the viewing angle and height of the FINESSE pointing mirror above the water surface. For these measurements the path length was 32 cm. The simulations were carried out at a resolution of 0.01 cm$^{-1}$ and then apodised with the FINESSE instrument lineshape which is described in Murray et al. (2023). The average simulated and apodised transmission is shown in figure 6a. Despite the short path length, absorption can be seen by $CO_2$ at 667 cm$^{-1}$ and water vapour below 700 cm$^{-1}$ and above 1300 cm$^{-1}$ .

The surface temperature and then emissivity were retrieved for each water view using equations 7 and 5 respectively. The individual emissivity retrievals were then averaged across all water views. This averaged emissivity is shown at full spectral resolution in figure 6b and is compared to the emissivity calculated using the refractive indices shown in figure 2. There is good agreement between the retrieved emissivity and the emissivity predicted using the Fresnel equations. The differences in predicted emissivity due to the different sets of refractive indices are smaller than the scatter in the retrieved emissivity. Below 500 cm$^{-1}$ and above 1300 cm$^{-1}$ the scatter of the retrieved emissivity increases. As emissivity is a smooth function of wavenumber, this scatter indicates poorer performance of the retrieval. The signature of the $CO_2$ band at 667 cm$^{-1}$ is also visible in the emissivity retrieval. These are regions of the spectrum where the transmission is lower (figure 6a) and there is less contrast between the up- and downwelling radiance measurements (figure 5a) so it is intuitive that the emissivity retrieval would perform less well in these conditions.

The uncertainty in the retrieved emissivity was then estimated. This was done by perturbing each of the input parameters of equation 5 individually and then re-running the emissivity retrieval. Table 1 lists the size and origin of the perturbations applied to each input parameter. The difference between the perturbed emissivity and the original emissivity was then calculated to give the error in the retrieved emissivity. The error in the retrieved emissivity was spectrally averaged into bins of 10 cm$^{-1}$ width and is shown in figures 7a and b. The retrieved emissivity has the lowest error in the atmospheric window between 800 and 1200 cm$^{-1}$ . In this region the largest contribution to the emissivity uncertainty is uncertainty in the surface temperature retrieval, followed by the effect of the NESR on the upwelling spectrum. Below 800 and above 1200 cm$^{-1}$ the uncertainty in the retrieved emissivity increases for all sources of uncertainty. Intuitively this can be thought of as a consequence of the reduced contrast between the up- and downwelling spectra that are measured by FINESSE and this can also be seen by examining equation 5. The NESR dominates the uncertainty below 550 cm$^{-1}$ and above 1350 cm$^{-1}$ suggesting that the error in future retrievals could be reduced further in these spectral regions by extending the measurement time.

The total uncertainty for the emissivity retrieval was calculated by summing all sources of error in quadrature. Figure 7c shows the final emissivity retrieval and associated error averaged in bins of 10 cm$^{-1}$ width. The solid lines show the theoretical emissivity calculated using the refractive indices from in figure 2. There is good agreement between the predicted and retrieved emissivity below 1400 cm$^{-1}$ which gives confidence in our emissivity retrieval technique.

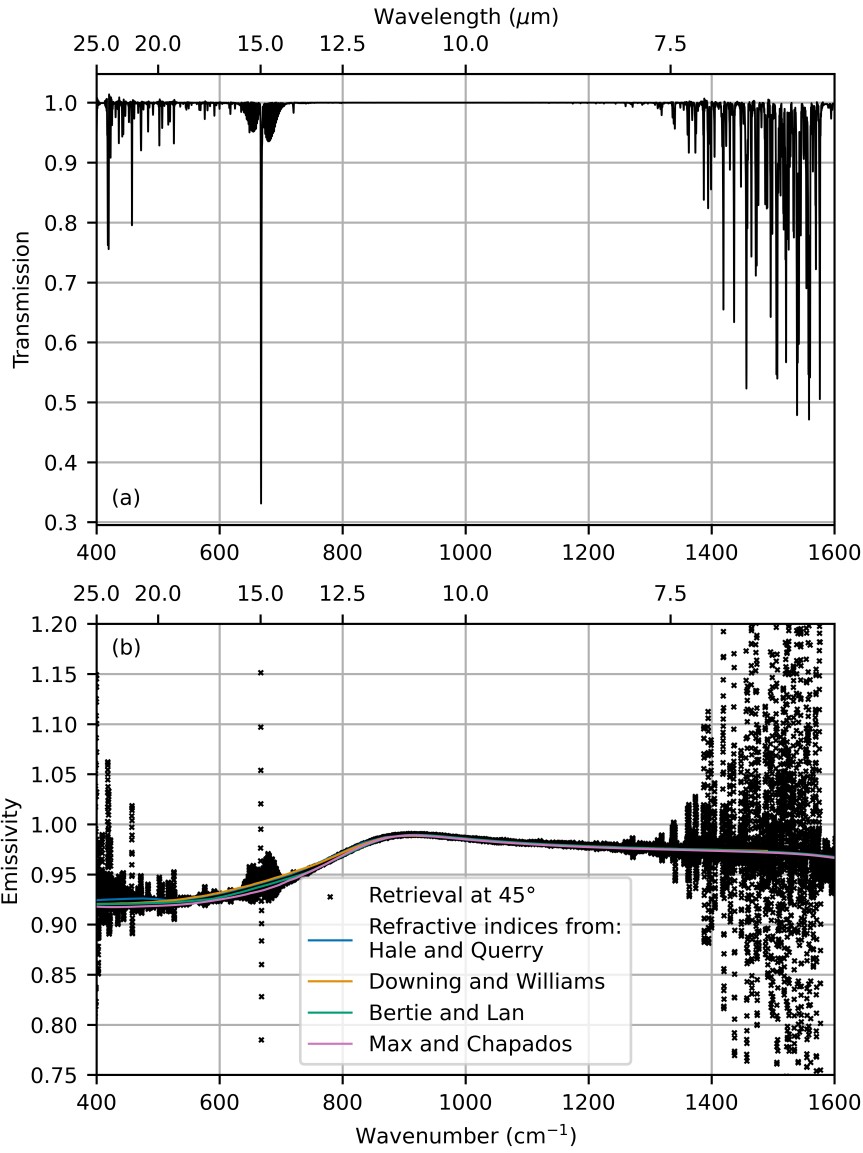

**Figure 6.** (a) Average transmission between the surface and FINESSE calculated using LBLRTM and apodised using the FINESSE instrument lineshape. (b) Retrieved emissivity at 45° and the theoretical emissivity calculated using the refractive indices shown in figure 2



**Table 1.** Perturbations applied to each of the input parameters in equation 5.

| Input Parameter | Perturbation Cause | Perturbation Value |
|---|---|---|
| Upwelling Radiance Spectrum | Temperature uncertainty of blackbody calibration targets | See figure 5b |
| Upwelling Radiance Spectrum | NESR | See figure 5b |
| Downwelling Radiance Spectrum | Temperature uncertainty of blackbody calibration targets | See figure 5c |
| Downwelling Radiance Spectrum | NESR | See figure 5c |
| Up- and Downwelling Radiance Spectra | Uncertainty in the emissivity of blackbody calibration targets | See figure 5b and c |
| Transmission | Uncertainty in the simulation of the intervening atmosphere caused by uncertainty in the pressure, temperature, humidity and $CO_2$ measurements | Accuracy of Vaisala PTU300 and GMP343<br><br>– 0.15 hPa<br><br>– 0.3 K<br><br>– 1% + 0.008 x reading<br><br>– 3 ppm + 0.01 x reading |
| Surface Temperature | Uncertainty of surface temperature retrieval even under ideal conditions | 0.025 K (Warwick, 2022) |



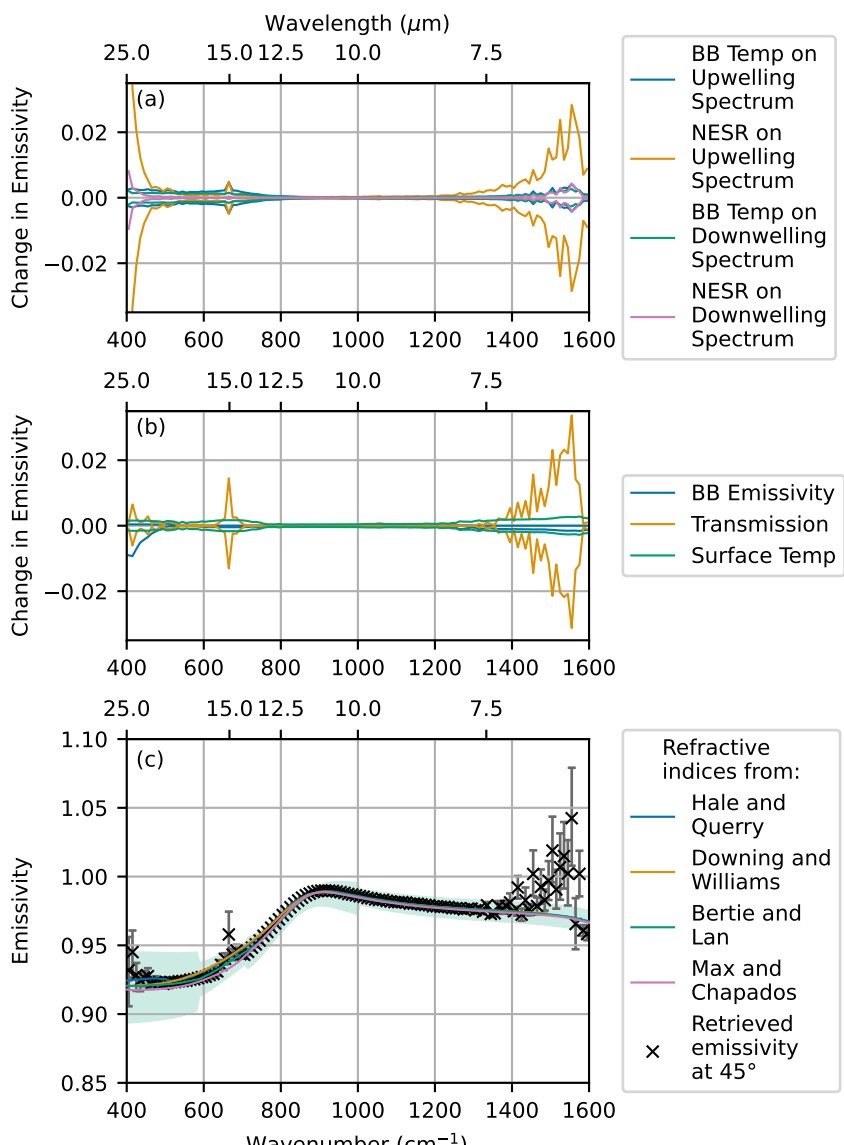

**Figure 7.** (a and b) Contributions to the uncertainty in the water emissivity retrieval. BB is short for blackbody. (c) retrieved emissivity at 45° with total error compared to emissivity modelled using the refractive indices from figure 2. The blue shading represents the uncertainty in the predicted emissivity due to uncertainty in the Bertie and Lan (1996) refractive indices.





## 4.2   Emissivity retrieval 17[th] March - angular dependence

Figure 8 shows the average upwelling and downwelling radiance for the three viewing angles measured. As the viewing angle of the downwelling radiation decreases from 130° to 110°, the measured downwelling radiance increases. This is due to the increased path length through the atmosphere. The increase in downwelling radiation is particularly noticeable in the far-infrared micro-windows between 500 and 600 cm$^{-1}$ . The upwelling radiation shows more noticeable atmospheric features as the upwelling viewing angle increases. This could either be because the emissivity decreases as the viewing angle increases leading to a larger proportion of the downwelling radiation being reflected into the instrument field of view, or because more noticeable atmospheric features are caused by the longer path length between the surface and the instrument. The measurements taken on 17[th] March differ from those taken on 11[th] February because for these measurements the water was at ambient temperature, rather than heated. This difference can be seen in the reduced contrast between up- and downwelling spectra, particularly in the far-infrared.

The emissivity of water at the three viewing angles was retrieved in the same manner as in section 4.1 and is shown at full spectral resolution by the grey points in figure 9. This is compared to the emissivity calculated using the refractive indices shown in figure 2. There is good agreement between the predicted and retrieved emissivity between 750 and 1250 cm$^{-1}$ . Below 750 cm$^{-1}$ and above 1250 cm$^{-1}$ the scatter in the retrieved emissivity increases indicating that the emissivity retrieval is less successful in these regions. Additionally, some retrieved emissivity values are nonphysical. This highlights the difficulty of retrieving surface emissivity when there is low contrast between the up- and downwelling radiation. We therefore applied a filter to target the regions in the far-infrared where there was good contrast between the up- and downwelling radiation by retrieving the emissivity in regions where the upwelling radiation was at least 3 mW m$^{-2}$ sr$^{-1}$ (cm$^{-1}$)$^{-1}$ higher than the downwelling radiation. In the far-infrared this has the consequence of retrieving the emissivity in the far-infrared micro-windows. The cut-off value of 3 mW m$^{-2}$ sr$^{-1}$ (cm$^{-1}$)$^{-1}$ was chosen as a compromise between retaining data in the far-infrared and removing nonphysical values. The effect of this filtering is shown in figure 9 by the black points. We retain this approach in the following analysis.

The uncertainty in the retrieved emissivity was calculated in the same way as in section 4.1 and then filtered and averaged spectrally in 10 cm$^{-1}$ bins. The contributions to the emissivity uncertainty for the 50° viewing angle are shown in figure 10a and b. Similar to the uncertainty for the February measurements (figure 6), the uncertainty in the retrieved emissivity increases below 800 and above 1200 cm$^{-1}$ . The dominant contributors to the uncertainty are still the surface temperature retrieval and the NESR.

Figure 10c shows the final emissivity at the three viewing angles. The solid lines are the emissivity predicted using the refractive indices shown in figure 2 and the blue shading is the uncertainty in the predicted emissivity based on uncertainty in the Bertie and Lan (1996) refractive indices. Throughout the spectrum, including in the far-infrared, there is good agreement between the predicted emissivity value and the retrieved value and a clear dependence of the emissivity on viewing angle. The uncertainty in the predicted emissivity caused by uncertainty in the Bertie and Lan (1996) refractive indices is generally

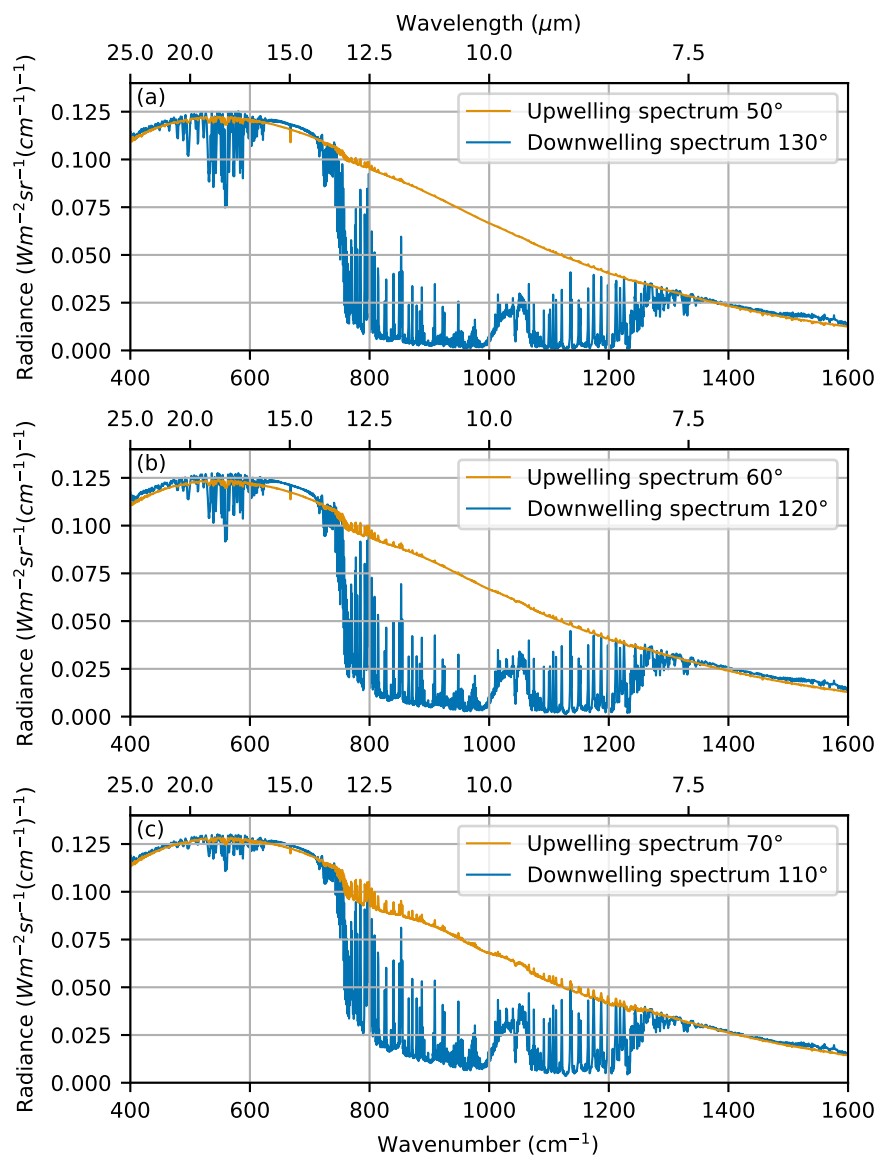

**Figure 8.** Average up- and downwelling radiance for the (a) 50° (b) 60° and (c) 70° measurements.

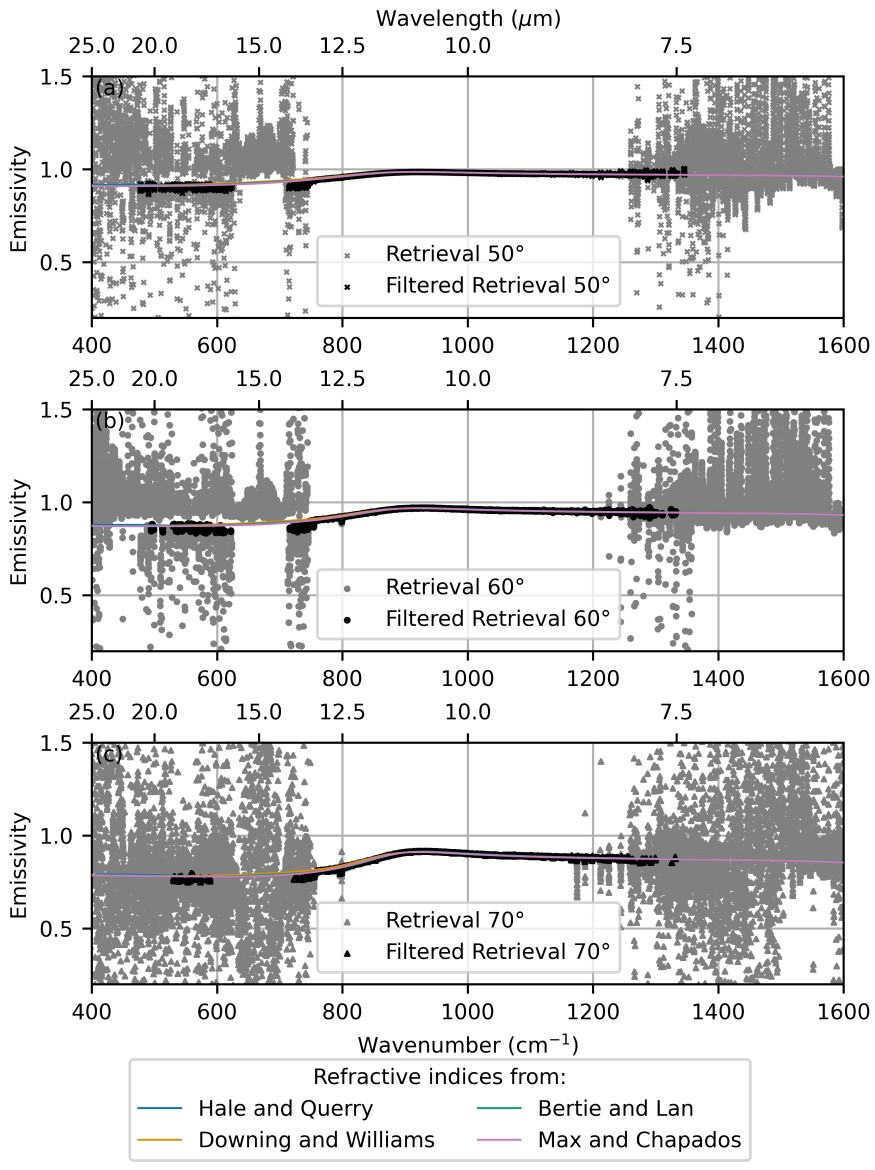

**Figure 9.** Emissivity retrievals at (a) 50° (b) 60° and (c) 70°. The solid lines are the theoretical emissivity calculated using the refractive indices shown in figure 2, the differences in predicted emissivity are not easy to distinguishable at this scale. The grey points are the emissivity retrieval at full resolution and the back points are the emissivity retrieval once the filter has been applied.

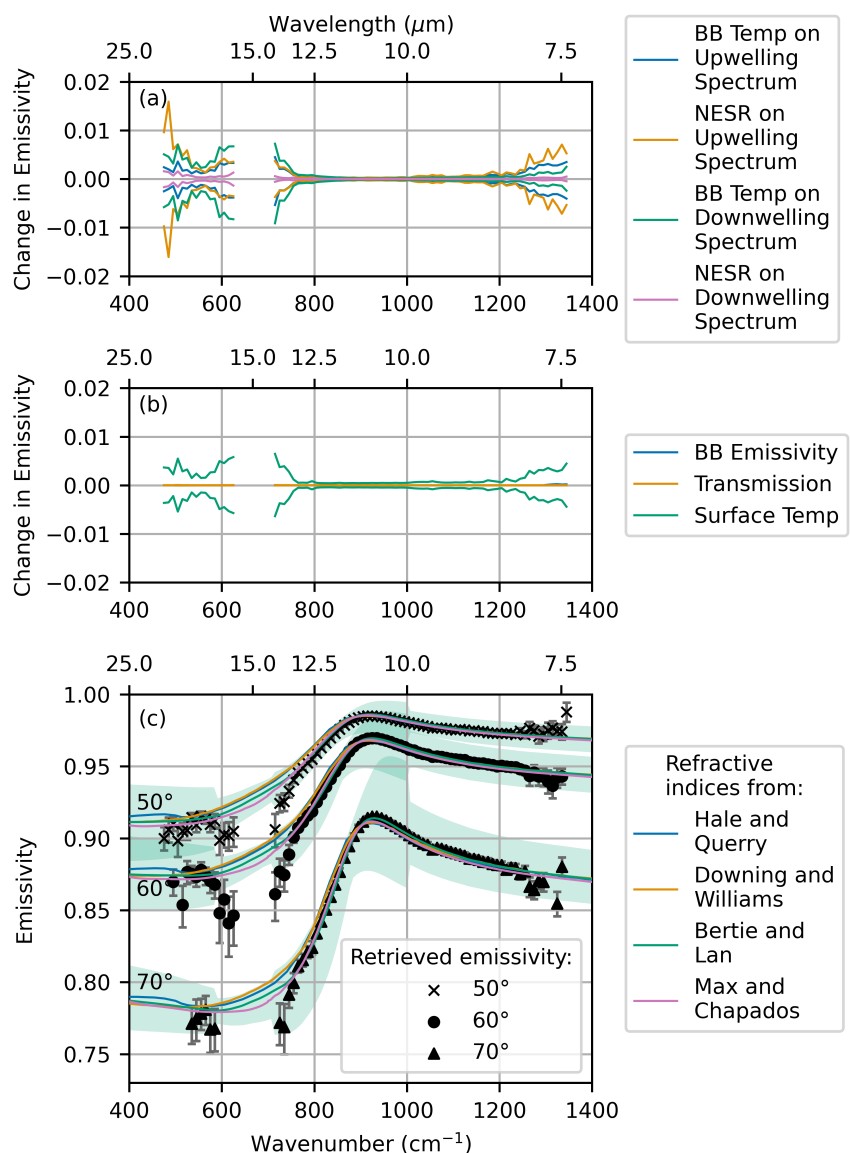

**Figure 10.** (a and b) Contributions to the uncertainty in the water emissivity retrieval at an angle of 50°. (c) (black) Retrieved emissivity with error bars showing the total error compared to the emissivity modelled using the refractive indices from figure 2.





larger than the uncertainty in the retrieved emissivity values, suggesting that the measured emissivity values could be used as
an independent constraint on the values of the refractive indices.

## 5 Conclusions

We have demonstrated a method for retrieving surface emissivity in the far- and mid-infrared using novel in-situ radiance measurements from the FINESSE instrument. Two sets of measurements of the emissivity of distilled water were made from the rooftop of Imperial College London. The first set of measurements was of heated water at an angle of 45°. These measurements successfully demonstrate our emissivity retrieval method and provide the first published measurements of water emissivity below 667 cm$^{-1}$ . The second set of measurements were made at angles of 50°, 60° and 70° to observe the angular variation in the emissivity of unheated water. The emissivity was successfully retrieved in the far- and mid-infrared although the scatter in the retrieved emissivity values was larger in the far-infrared and some nonphysical values were retrieved. This is because of the increased complexity of the retrieval in the far-infrared due to the absorption and emission of the atmospheric layer between the surface and the instrument and the decreased contrast between the up- and downwelling radiation compared to the heated water case. To circumvent these issues we developed a filter based on the contrast between the up- and downwelling radiance values. While the application of this filter reduces the wavenumber range over which the emissivity can be determined, the remaining retrievals show angular and spectral behaviour that is consistent with predicted emissivities using Fresnel equations and available refractive index compilations, giving confidence in the measurement quality and retrieval approach.

Analysis of uncertainty sources shows that the uncertainty increases at wavelengths below 750 cm$^{-1}$ and above 1250 cm$^{-1}$. The magnitude of the uncertainty is dependent on the measurement conditions, however on both days of measurement the largest source of uncertainty was the FINESSE NESR. Our retrievals match the theoretical simulations within quoted uncertainties across the majority of the 400 to 1400 cm$^{-1}$ spectral range, the shape and magnitude appears more consistent with the values from Max and Chapados (2009) at wavenumbers below approximately 900 cm$^{-1}$ . Conversely, the fit is slightly better to Bertie and Lan (1996) at higher wavenumbers. The observational uncertainties across much of the range, but particularly between 750 and 1250 cm$^{-1}$ are significantly smaller than those derived from uncertainties in the refractive indices, implying that the FINESSE retrievals could be used to provide a tighter constraint in this region.

These retrievals of water emissivity in the far- and mid-infrared demonstrate the potential of FINESSE to measure the emissivity of surfaces that are more difficult to model such as snow. Accurate measurements of the emissivity of snow and ice surfaces in the far-infrared in conjunction with characterisation of the snow micro-physical properties are needed to support the FORUM and PREFIRE satellite missions as well as to further improve global climate models. Such measurements have recently been taken during a deployment of FINESSE to Andøya, Norway and these will be documented in a future publication.

*Data availability.* The calibrated radiance data and auxiliary atmospheric measurements are available online at DOI 10.5281/zenodo.10377874



*Author contributions.* The study was conceptualised by HB and JM. LW and JM carried out the experiments. LW performed the data analysis
using methods developed jointly with JM. HB was responsible for supervision and funding acquisition. This manuscript was written by LW
with editing input from JM and HB.

*Competing interests.* The authors declare that they have no conflict of interest.

*Acknowledgements.* During the measurements and data analysis LW was funded by a CASE partnership between EPSRC and the National
Physical Laboratory (Grant No.EP/R513052/1). HB and JM were funded as part of NERC's support of the National Centre for Earth Obser-
vation under Grant No. NE/R016518/1.



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
