# Peer review of "The Far-INfrarEd Spectrometer for Surface Emissivity (FINESSE) Part II: First measurements of the emissivity of water in the far-infrared."

_Atmospheric Measurement Techniques, 2024_

## Author Comment (AC1)

*Figure A. (a) the downwelling radiance measurements made during two consecutive measurement cycles on 11ᵗʰ February. (b) the difference between these spectra and their average showing that the difference is smaller than the total measurement error.*

---

## Author Response (AR1)

**Responses to reviewer's comments**

**RC1**

Thank you for the comments on this work. We would like to clarify that the measurements described in this paper were taken outside in-situ rather than in a laboratory environment. The choice of distilled water was made to enable a straightforward comparison to Fresnel theory using (as you indicate) relatively well-known refractive indices for water. This allows an assessment of the quality of the measurements and retrieval approach. However, the intended purpose of the method described here is ultimately to make in-situ measurements of the far-infrared emissivity of surfaces that are more difficult to model. Although out of scope for this paper, since performing this work FINESSE has been deployed overseas and a future publication will show how its measurements have enabled the emissivity of snow and ice to be derived in the field.

Please find the responses to the comments below:

*The information and performance of the instrument, FINESSE, should be documented to provide more and sufficient details, despite of the cited paper (Murray 2023), which by the way was not accessible during the review. This includes how the NESR is determined, whether it differs with regard to down- vs. up-welling radiance measurements, and whether it varies with environment conditions (temperature, humidity, etc.). These are critical to the in-situ applications, which is proposed here for the method.*

The part I of this paper is available as a preprint here: https://doi.org/10.5194/amt-2024-22 It is intended that the papers be published together and that the interested reader can find the technical details of the calibration and "level 1" products in Part 1. We will contact the journal to ask that the papers be clearly linked. In this paper we use NESR to refer to the spectrally uncorrelated detector noise. This is a property of the detector and not related to the external conditions. We have found the NESR to be stable across a variety of lab and in-situ environments. We have added the following to the paper on line 168 as a quick description of how the NESR was calculated:

"The NESR is associated with the instrument detector noise and is spectrally uncorrelated. The NESR can be calculated from the difference of consecutive calibrated radiance spectra on the assumption of an unchanging scene. For a fixed instrument configuration, throughput, resolution, and acquisition time, the NESR is scene independent, however, being spectrally uncorrelated it can be reduced by averaging spectra. The radiance uncertainty on the calibrated spectra, due to the temperature and emissivity uncertainties of the blackbody cavities was calculated for each spectrum, this uncertainty is scene dependent and spectrally correlated, it cannot be reduced by averaging. Details of the uncertainty determination can be found in part I of this paper (Murray et al., 2024)."

*Line 10: "first published retrieval". I found this claim of "first" in need of justification or modification. Many studies measured the water emissivity down to the FIR, such as Downing & Williams (1975). Although the results were presented in the form of refractive index, they are essentially the same information and were obtained similarly in laboratory settings. Other statements related: Line 127: "no published values ...".*

Thank you for this comment. We completely agree that material refractive indices are strongly coupled to the emissivity of that material but they are not the same. In the paper we refer to several sets of refractive index compilations. These compilations use data from a variety of experiments and some measurements are used in multiple compilations. The Downing & Williams (1975) paper makes use of several sources of data in the far-infrared region including transmission measurements (Robertson & Williams 1971, Robertson et al. 1973) and reflectance measurements (Rusk et al. 1971). The methodology for the transmission measurements is significantly different from our approach. The reflectance measurements of Rusk et al. differ from our work as we simultaneously determine the surface temperature and emissivity, and our measurements are made in-situ rather than in a tightly controlled laboratory environment.

To clarify this we have altered line 10 to read "these retrievals are, to the best of our knowledge, the first published simultaneous retrievals of the surface temperature and emissivity of water that extend into the far-infrared"

We have also altered line 129 to read "To the best of our knowledge there are no published simultaneous retrievals of the emissivity and surface temperature of distilled, fresh or sea-water that extend into the far-infrared."

Finally, we have altered line 249 in the conclusion to read "These measurements successfully demonstrate our emissivity retrieval method and provide the first joint retrieval of surface emissivity and surface temperature of water in the far-infrared."

*Method (eq. 1). Some key assumptions are not explicitly stated and sufficiently discussed. For example, how well can specular reflection (neglect of photons from different directions) be assumed? Caption of Fig 4 stated "as flat as possible": how flat is water surface? It is desirable to more rigorously test this condition (specular reflectance). Moreover, can sky condition be assumed homogeneous (in the lab setting in this experiment and possibly more complicated when applied in situ)? And is the neglect of angular dependence of reflectance (emissivity) - a good one too? Note this may be especially problematic in the real retrieval environments, e.g. due to the surface waves. How would you detect, avoid or mitigate these issues in the in-situ retrievals?*

These measurements were made on clear sky days with stable atmospheric conditions and very little wind. The nearby Met Office weather station at Battersea Heliport recorded wind speeds of 1 m/s on the first measurement day and 2 m/s on the second measurement day (Met Office 2023) and we further shielded the water surface from the wind using the screen shown in figure 4. This meant that the water surface was flat and level as nothing was disturbing it. We have changed the ambiguous wording in the caption of figure 4 on page 9 to read "The green screen in figure (b) was used to shield the water surface from the breeze, again to ensure the surface was flat."

We note that Newman et al. (2005) also found the specular assumption matched their ground-based measurements and reported no measurable difference in emissivity caused by the ripples caused by the heating action of their ultrasound bath. Any ripples caused by the wind in our set up were far smaller than those caused by such a heating action, therefore we are confident that the specular assumption is reasonable for the set-up described in this paper. For retrievals of other surfaces such as snow, different assumptions are required. For example, in

the past, it has been assumed that snow acts as a Lambertian reflector (e.g. Hori et al. 2006, Bellisario et al. 2017). This is something that we intend to explore in future work.

On line 1 in the abstract, we have clarified that the method described is for "retrieving the surface emissivity of specular surfaces". We have also altered the conclusion on line 246 to read "We have demonstrated a method for retrieving surface temperature and surface emissivity of specular surfaces in the far- and mid-infrared using novel in-situ radiance measurements from the FINESSE instrument."

Our measurements do rely on the downwelling radiation being homogeneous and consistent throughout the measurements. The homogeneity clause is met in-situ by carrying out measurements only on days with no cloud cover. We have added this stipulation to the paper on line 143: "Clear sky conditions were selected to ensure the homogeneity and stability of the downwelling radiance over the course of the measurements".

As part of the analysis of the results, the stability of the downwelling radiance with time is assessed. This was not included in the paper for reasons of brevity however figure A below shows an example of the analysis that we carry out. This figure shows that the difference between the downwelling spectra recorded on two consecutive measurement cycles (roughly 15 minutes apart) is smaller than the measurement error on the spectrum. This gives us confidence that the downwelling radiance is not changing over the course of the emissivity measurement. We have added the following to line 165 of the paper:

"The variability of the downwelling spectra was also plotted (not shown), this confirmed that the downwelling radiance was consistent during the emissivity retrieval."

[Figure]

*Figure A. (a) the downwelling radiance measurements made during two consecutive measurement cycles on 11$^{th}$ February. (b) the difference between these spectra and their average showing that the difference is smaller than the total measurement error.*

The emissivity of a sea surface can be affected by the presence of surface waves particularly at more oblique viewing angles and in the case of higher windspeeds (e.g. Masuda et al. 1988). We have no plans to measure at sea using this set-up as not only would we need to consider these wave and angular effects, but also find a means to compensate for ship movement and develop a strategy to determine the representative angle for the downwelling radiation.

*By the way, the method, cited as Newman 2005 here, should probably acknowledge earlier, more original works, e.g., those behind the Marine AERI developments.*

Thank you for the suggestion of adding additional earlier sources describing the emissivity retrieval method. The method of retrieving incident viewing angle described by Smith et al. (1996) is of particular interest as we develop our retrieval process for the measurement of non-specular surfaces.

We have changed line 69 to read "We have chosen to adapt an emissivity retrieval method that has been successfully applied for the retrieval of surface emissivity in the mid-infrared using high resolution spectra (Newman et al., 2005; Fielder and Bakan, 1997; Smith et al., 1996). This method has been used to retrieve surface emissivity of water at different temperatures and

salinities using observations from several mid-infrared interferometers in the lab, from aircraft and in-situ from data taken from an oceanographic cruise. The method has recently been adapted for the retrieval of far-infrared snow and ice emissivity from aircraft measurements (Bellisario et al., 2017)."

*Line 232. In relation to the above comments, I found the proof of the method for in-situ application not sufficient enough. A more rigorous validation plan is necessary to prove the retrievals, especially in non-lab environments. Otherwise, the claims should be limited to what is done (lab as opposed to in situ).*

We hope that the comments and changes to the paper mentioned above have clarified the point that this was an in-situ rather than laboratory study. We therefore think that these measurements show the capability of the instrument and technique. We have added the following to the paper on line 268 to highlight the point that the method will need to be extended for the measurement of non-specular surfaces:

"These retrievals of water emissivity in the far- and mid-infrared demonstrate the potential of FINESSE. Upcoming work will develop this method further for application to surfaces that are more difficult to model such as snow."

As noted in our earlier response, plans for this are already underway.

*Possibly trivial is that the measurement is about water, as opposed to atmosphere, properties. A note may be necessary to explain why the work fits AMT.*

We have added the following to line 269 of the paper "Accurate measurements of the surface emissivity in the far-infrared are needed to support both the surface and atmospheric retrievals from the FORUM and PREFIRE satellite missions as well as to further improve global climate models."

**RC2**

Thank you very much for your comments, we are glad that you find the paper novel, relevant and clearly explained.

*Minor Comments:*

*Figure 1 is an excellent schematic.*

*Table 1 and Figure 7 are a very important aspect of this work.  My minor recommendation would be to further motivate the reasoning for the 0.025 K uncertainty in the surface temperature (i.e. add a sentence in the manuscript), especially because you point it out as a significant contributor in certain spectral regions. I tried to find the referenced PhD thesis but could not find it online after a short search. This value seemed markedly small to me, though I have limited experience in retrieving surface temperature. Any uncertainty value is acceptable, so long as it is listed as an assumption in the paper (which it clearly is in Table 1 already).*

Thank you for the comment, from this we can see that the wording of this term in the table is misleading. What is meant here is the precision at which the surface temperature can be determined using the computational method described in the paper. This is not the final uncertainty on the surface temperature retrieved by FINESSE. This final surface temperature uncertainty is larger and is influenced by all the contributions listed in table 1. To clarify this, we have changed the wording in Table 1 to read "Precision of the computational method used to

retrieve surface temperature" and edited line 194 to read "The uncertainty in the retrieved emissivity was then estimated. This was done by perturbing each of the input parameters of equation 5 individually and then re-running the surface temperature and emissivity retrieval." This makes it clear that the surface temperature was re-retrieved when each perturbation was applied.

We have added the DOI of the referenced PhD thesis (https://doi.org/10.25560/104120) to the references.

*Technical Corrections:*

*Line 58: "in provided in Part 1" should be "is provided in Part 1"*

Thank you, this has been corrected.

**References**

Bellisario, C., Brindley, H. E., Murray, J. E., Last, A., Pickering, J., Harlow, R. C., et al. (2017). Retrievals of the far infrared surface emissivity over the Greenland Plateau using the Tropospheric Airborne Fourier Transform Spectrometer (TAFTS). *Journal of Geophysical Research: Atmospheres*, 122, 12,152–12,166, doi:10.1002/2017JD027328

Downing, H. D., & Williams, D. (1975) Optical constants of water in the infrared, *J. Geophys. Res.*, 80(12), 1656–1661, doi:10.1029/JC080i012p01656.

Hori, M., Aoki, T., Tanikawa, T., Motoyoshi, H., Hachikubo, A., Sugiura, K., Yasunari, T. J., Eide, H., Storvold, R., Nakajima, Y., & Takahashi, F. (2006) In-situ measured spectral directional emissivity of snow and ice in the 8–14 µm atmospheric window, *Remote Sensing of Environment*, 100, 486–502, doi:10.1016/j.rse.2005.11.001, 2006

Masuda, K., Takashima, T., & Takayama, Y. (1988) Emissivity of Pure and Sea Waters for the Model Sea Surface in the Infrared Window Regions. *Remote Sensing of Environment*, 24, 313-329, doi:10.1016/0034-4257(88)90032-6

Met Office (2023) MIDAS Open: UK mean wind data, v202308. NERC EDS Centre for Environmental Data Analysis, 03 October 2023. doi:10.5285/68920a29caf44f21be6371d9f87f578b. https://dx.doi.org/10.5285/68920a29caf44f21be6371d9f87f578b

Newman, S. M., Smith, J. A., Glew, M. D., Rogers, S. M., & Taylor, J. P. (2005) Temperature and salinity dependence of sea surface emissivity in the thermal infrared, *Quarterly Journal of the Royal Meteorological Society*, 131, 2539–2557, doi:10.1256/qj.04.150

Robertson, C.W., & Williams, D. (1971) Lambert Absorption Coefficients of Water in the Infrared*, *J. Opt. Soc. Am*. 61, 1316-1320, https://opg.optica.org/josa/abstract.cfm?URI=josa-61-10-1316

Robertson, C.W., Curnutte, B. & Williams, D. (1973) The infra-red spectrum of water, *Molecular Physics*, 26:1, 183-191, doi:10.1080/00268977300101501

Rusk, A., Williams, D., & Querry, M. (1971) Optical Constants of Water in the Infrared*, *J. Opt. Soc. Am*.  61, 895-903, doi:10.1364/JOSA.61.000895

Smith, W., Knuteson, R., Revercomb, H., Feltz, W., Howell, H., Menzel, W., Nalli, N., Brown, O., Brown, J., Minnett, P. and McKeown, W. (1996). Observations of the Infrared Radiative Properties of the Ocean–Implications for the Measurement of Sea Surface Temperature via Satellite Remote Sensing. *Bulletin of the American Meteorological Society* 77(1), 41-52, doi: 10.1175/1520-0477(1996)077%3C0041:OOTIRP%3E2.0.CO;2